# Health services costs for lung cancer care in Australia: Estimates from the 45 and Up Study

**David E. Goldsbury**[1]*, **Marianne F. Weber**[1,2], **Sarsha Yap**[1], **Nicole M. Rankin**[1,3], **Preston Ngo**[1,2], **Lennert Veerman**[1,2,4], **Emily Banks**[5], **Karen Canfell**[1,2,6], **Dianne L. O'Connell**[1,2,7]

**1** Cancer Research Division, Cancer Council NSW, Sydney, NSW, Australia, **2** Sydney School of Public Health, The University of Sydney, Sydney, NSW, Australia, **3** Faculty of Medicine and Health, The University of Sydney, Sydney, NSW, Australia, **4** School of Medicine, Griffith University, Southport, Queensland, Australia, **5** National Centre for Epidemiology and Population Health, Australian National University, Canberra, ACT, Australia, **6** Prince of Wales Clinical School, UNSW Medicine, Sydney, NSW, Australia, **7** School of Medicine and Public Health, University of Newcastle, Newcastle, NSW, Australia

* davidg@nswcc.org.au

## Abstract

### Background

Of all cancer types, healthcare for lung cancer is the third most costly in Australia, but there is little detailed information about these costs. Our aim was to provide detailed population-based estimates of health system costs for lung cancer care, as a benchmark prior to wider availability of targeted therapies/immunotherapy and to inform cost-effectiveness analyses of lung cancer screening and other interventions in Australia.

### Methods

Health system costs were estimated for incident lung cancers in the Australian 45 and Up Study cohort, diagnosed between recruitment (2006–2009) and 2013. Costs to June 2016 included services reimbursed via the Medicare Benefits Schedule, medicines reimbursed via the Pharmaceutical Benefits Scheme, inpatient hospitalisations, and emergency department presentations. Costs for cases and matched, cancer-free controls were compared to derive excess costs of care. Costs were disaggregated by patient and tumour characteristics. Data for more recent cases identified in hospital records provided preliminary information on targeted therapy/immunotherapy.

### Results

994 eligible cases were diagnosed with lung cancer 2006–2013; 51% and 74% died within one and three years respectively. Excess costs from one-year pre-diagnosis to three years post-diagnosis averaged ~$51,900 per case. Observed costs were higher for cases diagnosed at age 45–59 ($67,700) or 60–69 ($63,500), and lower for cases aged ≥80 ($29,500) and those with unspecified histology ($31,700) or unknown stage ($36,500). Factors associated with lower costs generally related to shorter survival: older age (p<0.0001), smoking (p<0.0001) and unknown stage (p = 0.002). There was no evidence of differences by year of

**Data Availability Statement:** The data cannot be made available by the authors. These are third party data not owned or collected by the authors and on-provision is not permitted by the relevant

data custodians (Sax Institute, Services Australia, NSW Ministry of Health), as it would compromise the participants' confidentiality and privacy. The data contain potentially identifying and sensitive patient information. However the data are available from the data custodians for approved research projects - data access enquiries can be made to the Sax Institute (see https://www.saxinstitute.org.au/our-work/45-upstudy/governance/ for details). Other researchers would be able to access these data using the same process followed by the authors.

**Funding:** The authors received no specific funding for this work. EB and KC receive salary support from the National Health and Medical Research Council of Australia (PRF #1136128 and CDF #1082989 respectively). KC is an investigator on an unrelated investigator-initiated trial of cytology and primary HPV screening in Australia (Compass), which is conducted and funded by the Victorian Cytology Service (VCS), a government-funded health promotion charity. The VCS has received equipment and a funding contribution for the Compass trial from Roche Molecular Systems and Ventana. However, neither the authors nor the authors' organisations receive direct funding from industry for this trial or any other project. No funder had a role in study design, data collection and analysis, decision to publish, or preparation of the manuscript.

**Competing interests:** The authors have declared that no competing interests exist.

diagnosis or sex (both $p$>0.50). For 465 cases diagnosed 2014–2015, 29% had subsidised molecular testing for targeted therapy/immunotherapy and 4% had subsidised targeted therapies.

## Conclusions

Lung cancer healthcare costs are strongly associated with survival-related factors. Costs appeared stable over the period 2006–2013. This study provides a framework for evaluating the health/economic impact of introducing lung cancer screening and other interventions in Australia.

## Introduction

Lung cancer is among the most common cancers worldwide [1]. In Australia it is the fifth most commonly diagnosed cancer and the most common cause of cancer death, with an estimated 12,800 new cases diagnosed and 9,000 deaths in 2019 [2]. Lung cancer care accounts for a substantial proportion of Australian government healthcare costs [3, 4]. The poor prognosis and healthcare resource requirements mean there is a need for cost-effective interventions aimed at primary prevention, screening, early detection, and improvements in treatment. Costing studies are key to informing cost-effectiveness evaluations, informing policy decisions for cancer control and improving public health. While health services costs are setting-specific, the cost-effectiveness evaluations they inform can be translated across populations with similar healthcare systems by identifying the key drivers of effectiveness and cost-effectiveness.

Lung cancer care is largely guided by tumour histology and disease stage at diagnosis [5, 6]. Lung cancer is commonly diagnosed at an advanced stage, with over 40% of cases in Australia having distant metastases at diagnosis [7] and 50–60% in the UK, Canada and the US having advanced disease at diagnosis [8, 9]. Diagnosis at an earlier stage may be possible with more timely investigations and referrals [10] or with the widespread implementation of low dose CT screening of high-risk individuals in a screening program. However, any shift in the population-wide stage distribution would impact on patterns of treatment, survival, quality of life and healthcare costs [11].

Individual patient care and the related costs vary based on patient and tumour characteristics. To date, little has been reported on population-based costs for patient subgroups in Australia, in part due to a lack of available data. The Australian healthcare system includes government-funded universal coverage of many medical costs, with other costs mainly covered by private health insurance or patients' out-of-pocket costs. Previous Australian studies have assessed costs of treatment but did not include costs for other related aspects of cancer care, such as general practitioner or specialist consultations [3, 12, 13]. Studies in the UK, US, and New Zealand have reported the excess health system costs of care for lung cancer patients relative to cancer-free controls by age and sex [14–16]. Changes in costs over time can be quantified using this method in settings like Australia where there is a universal healthcare system with linked records across healthcare sectors or central recording of all encounters with the healthcare system, or where large health funds cover all costs of healthcare for a broad cross-section of the population. We previously used this method to report overall costs for each of the ten most common cancer types in Australia, including lung cancer [4], but did not disaggregate these for important patient subgroups.

To assess the potential impact of both existing and emerging lung cancer interventions on future healthcare costs, information is needed on the distribution of lung cancer care costs for key patient subgroups, referred to as the "usual care" scenario in cost-effectiveness analyses. For example, since 2013, a number of targeted therapies/immunotherapies (e.g. gefitinib, erlotinib, nivolumab) have been subsidised by the Australian government for the treatment of an expanding range of indications relating to lung cancer [17], and molecular testing to determine access to such therapies is increasingly performed. However, uptake was still relatively low during the period described in this study [18]. These new methods of cancer care have the potential to improve survival and quality of life, but they are expensive [19], and the broader impact of these new treatment technologies on the health system remains unclear. Detailed costs for lung cancer care are also timely given the policy setting in Australia and other high-income countries, with the demonstration of lung cancer screening effectiveness in two key trials [20, 21]. Stage-specific cost estimates related to lung cancer incidence (e.g. age, smoking status) and factors related to treatment decisions (e.g. histological type, accessibility of services) are needed to update cost-effectiveness evaluations of lung cancer screening in Australia [22–24].

Our aim, therefore, was to provide detailed population-based estimates of the health system costs for lung cancer care using linked patient-level data from New South Wales (NSW), Australia. We estimated lung cancer costs for several patient and tumour subgroups that are likely to be relevant to cost-effectiveness evaluations of lung cancer interventions prior to the wider introduction of new therapies and technologies. We analysed costs prior to diagnosis, for initial treatment, continuing care, and end-of-life care, providing a detailed resource for future healthcare planning. We also examined potential changes in practice related to use of targeted therapies for more recently diagnosed cases who were identified using hospital records.

## Materials and methods

### Data sources

The study sample was selected from participants in the Sax Institute's 45 and Up Study, a longitudinal study of 267,153 people in NSW aged ≥45 years. The study methods and cohort have been described previously [25]. Briefly, potential participants were a random sample from the Medicare enrolment database held by Services Australia (formerly the Department of Human Services), which has near complete coverage of the population. Rural residents and people aged ≥80 years were oversampled. Participants completed a postal questionnaire during 2006–2009 and consented to linkage of their questionnaire data to routinely collected health information.

The linked health records included reimbursements for government-subsidised prescription medicines in the Pharmaceutical Benefits Scheme (PBS) and subsidised outpatient and medical services and some in-hospital procedures covered by the Medicare Benefits Schedule (MBS). The Sax Institute linked these records for study participants using a unique identifier. The Centre for Health Record Linkage [26] linked other health records for study participants including inpatient care in all NSW hospitals from the Admitted Patient Data Collection (APDC), emergency presentations from the Emergency Department Data Collection (EDDC), statutory notifications of cancer except keratinocyte (non-melanoma skin) cancers from the NSW Cancer Registry (NSWCR) and death notifications from the NSW Registry of Births, Deaths and Marriages (RBDM) (Fig 1). All data were fully anonymised before being made available for analysis.

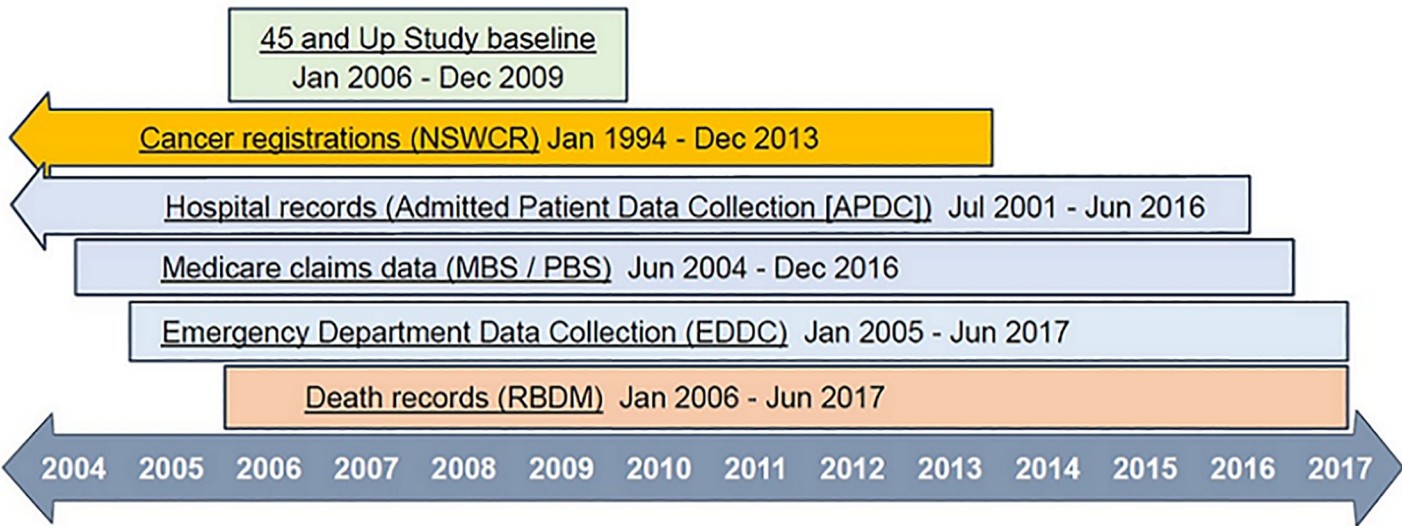

**Fig 1. Data sources and date coverage.** NSWCR: New South Wales Cancer Registry; MBS: Medicare Benefits Schedule; PBS: Pharmaceutical Benefits Scheme; RBDM: Registry of Births, Deaths and Marriages.

### Study sample

People included for this analysis were 45 and Up Study participants with a NSWCR-registered lung cancer, diagnosed during 2006–2013 after study recruitment. We excluded participants with:

i. probable linkage errors and those aged <45 years at baseline;

ii. invasive cancer diagnosed prior to study recruitment (NSWCR);

iii. any non-lung cancer diagnosed (except if first diagnosed with lung cancer) after recruitment;

iv. self-reported history of cancer (except keratinocyte cancers);

v. healthcare subsidised by the Australian Government Department of Veterans' Affairs (DVA), as their prescription medicine records were not available. DVA clients were identified via self-report or any mention of DVA coverage in hospital or ED records;

vi. unknown day/month of diagnosis, or first diagnosed after death.

The remaining cases were matched to controls who had no record of cancer at any time and who were alive on the date their matched case was diagnosed. Up to three controls were matched to each case (without replacement) based on age (±5 years), sex (female; male), Local Government Area of residence (~150 areas in NSW) and smoking status (never or quit >15 years; ex-smoker quit < = 15 years; current smoker) at recruitment. People with no information for the matching variables were excluded, as were cases with no matched controls (Fig 2). Lung cancer cases were identified using topography code C34 from the International Classification of Diseases, Tenth Revision (ICD10).

**Cases diagnosed after 2013.** To separately investigate the introduction of targeted therapies in real-world lung cancer care, we used hospital admission records to identify more recently diagnosed lung cancer cases in the 45 and Up Study cohort, with the sensitivity and positive predictive value of the method of diagnosis, compared to using NSWCR information,

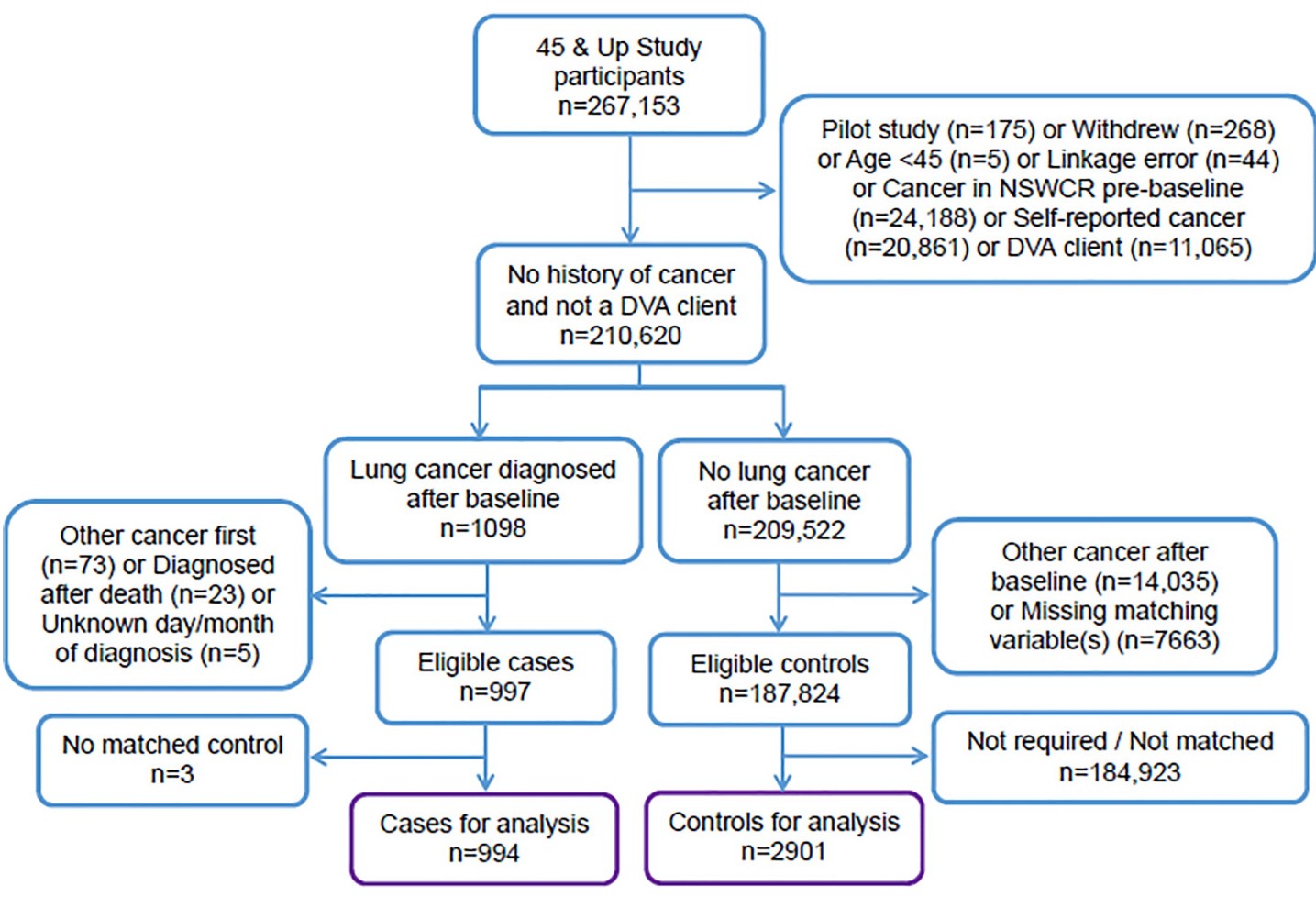

**Fig 2. Cohort selection flow diagram.** DVA: Department of Veterans' Affairs; NSWCR: New South Wales Cancer Registry.

both estimated at 80–85% [27, 28]. We identified lung cancer cases diagnosed during 2014–2015 who had no prior NSWCR lung cancer record, and investigated their health services use, also comparing with NSWCR cases diagnosed 2006–2013 (the matched cases from the main analysis). Histology and disease stage information was unavailable for these cases. We identified relevant PBS medicine records and MBS records relating to molecular testing for epidermal growth factor receptor (EGFR) or anaplastic lymphoma kinase (ALK), used to determine eligibility for certain targeted therapies [29].

### Definitions and estimation of health system costs

**Data for costs.** For this study we focused on direct health system (government-funded) costs. Costs were estimated from records of individuals' inpatient hospital episodes, ED presentations, and subsidised prescription medicines and medical services recorded in the PBS and MBS respectively. The period for which data on costs were available from all healthcare databases was 1 January 2005 to 30 June 2016 (Fig 1). This provided coverage of costs for a minimum of 2.5 years after diagnosis (controls were assigned the "diagnosis date" of their matched case), with median potential follow-up of 5.4 years after diagnosis.

Costs were estimated for time periods relative to the cases' diagnosis and death dates where applicable. If the time period of interest extended beyond the end of the study period then the matched case-control group did not contribute to the estimates for that specific period (e.g. if the case was diagnosed in December 2013 then 3 years post-diagnosis is after 30 June 2016). Costs for all related items and procedures were included.

The dates of supply of prescription medicines (PBS) and medical services (MBS) were used to assign costs to the relevant time period. Inpatient hospital costs were derived by linking the Australian Refined Diagnostic Related Group code for each hospitalisation to the average admission cost recorded in the 2010 National Hospital Cost Data Collection [30]. As hospitalisations could span multiple time periods, costs were apportioned across periods using the proportion of total length of stay falling in each period. For ED presentations, there was insufficient treatment information available to assign detailed costs so we used average costs from the 2010 National Hospital Cost Data Collection, assigned according to triage category and discharge status [30]. All cost values were converted to 2013 Australian dollars using the Australian Health Index from the Consumer Price Index [31].

**Statistical analysis.** The excess costs due to lung cancer were estimated for each case by taking their total healthcare costs and subtracting the average costs for their matched controls. If the case died then the included costs were censored at the date of death and the case-control group were excluded from calculations for subsequent time periods. If a control died, the matched case and their remaining control(s) were included in any subsequent estimates. Means of excess costs were calculated, along with the proportions of excess costs contributed by each source (inpatient hospitalisations; ED presentations; prescription medicines; other subsidised services in the MBS). Medians, standard deviations and inter-quartile ranges are reported in the Supporting Information (Table A in S1 File). Analysis was carried out using SAS v9.4 (SAS Inc., NC, US).

Cases were classified by NSWCR histological subtype: small cell lung cancer (SCLC); non-small cell lung cancer (NSCLC, comprising adenocarcinoma, squamous cell carcinoma and large cell carcinoma); other specified carcinoma; and other/unspecified [32]. Cases were also classified by NSWCR summary disease stage at diagnosis (localised; regional; distant metastases; unknown), sex, age at diagnosis (45–59 years; 60–69; 70–79; ≥80), year of diagnosis, and smoking status, remoteness of place of residence [33], socioeconomic disadvantage quintile for place of residence [34], and health insurance status (private health insurance; healthcare concession card; none) at recruitment. These factors were chosen due to their potential association with lung cancer incidence and treatment, as these are key factors relating to a potential lung cancer screening program and other potential interventions [22, 35]. Where possible, missing values were imputed from information for the same person recorded in other datasets. We also analysed cancer-related treatment (surgery, chemotherapy or radiotherapy; see Table B in S1 File) from the month of diagnosis onwards. We compared the distributions of patients' and disease characteristics in our cohort with the corresponding characteristics for all lung cancers in NSW or Australia. Cases were weighted to match the stage distribution for all lung cancers diagnosed in NSW 2010–2014.

**"Average case" costs.** To allow comparisons between patient subgroups defined by patients' and tumour characteristics, we calculated an "average case" excess cost for each case-control group. This used the excess costs from 1 year before to 3 years after diagnosis as previous work showed little excess costs >1 year before diagnosis for lung cancer cases [4], and 92% of cases had at least 3 years of potential follow-up after diagnosis. We summarised these costs by the factors described in the previous paragraph. To assess the independent effects of these factors, we also analysed the costs using gamma regression with a log-link [16], including all factors as covariates, with the effects considered statistically significant when p<0.05. A small

number of outlying excess costs were excluded from the regression analysis if they had a standardised Pearson residual <-4 or >5. Due to the potential for negative excess costs, all cost estimates were "shifted" by +$100,000 for this specific analysis so that all were >$0.

**Other cost estimates.** Monthly costs around the case's date of diagnosis and monthly costs at the end of life relative to the case's death date were estimated. As described previously [4], costs were also grouped into three phases of care: initial, continuing and terminal. For cases who:

- died before July 2016, the final year up to and including the death date defined the terminal phase.

- survived ≤1 year after diagnosis, the terminal phase started at the diagnosis date and no costs were attributed to other phases.

- survived at least two years, the first year after diagnosis defined the initial phase.

- survived >1 year but ≤2 years, the initial phase was the period from diagnosis until the start of the 12-month terminal phase.

- survived >2 years after diagnosis, the period between the end of the initial phase and the start of the terminal phase, or 30 June 2016 if the case was still alive, defined the continuing phase. Costs for this phase were estimated as an annual rate, excluding any case-control group with the continuing phase <3 months (1.6% of groups).

**Ethical approval.** The University of New South Wales Human Research Ethics Committee approved the conduct of the 45 and Up Study. The NSW Population and Health Services Research Ethics Committee (approval number 2014/08/551) approved this study.

## Results

### Study sample

There were 997 eligible cases diagnosed with lung cancer between recruitment and December 2013 (Fig 2). At least one matched control was identified for 994 cases (99.7%), with three matched controls for 938 (94%), two matched controls for 31 (3%) and one matched control for 25 cases (3%). The final study sample comprised 994 cases and 2901 controls, with a mean age of 71 years at the cases' diagnosis.

Compared with all lung cancers in NSW/Australia, the study cohort had a similar distribution for sex, age and disease stage at diagnosis, but a higher proportion of cases with adenocarcinoma and people from outside of major cities (in line with oversampling of rural areas in the study cohort) or from more disadvantaged areas (Table 1). Sixty-one percent of SCLC cases had distant metastases, compared with 47% of all other cases. Seventy-five percent of cases had a record of cancer-related treatment: 22% received surgery, 46% chemotherapy and 48% radiotherapy. Ninety percent of cases aged 45–59 years had a record of cancer-related treatment, compared with 83% of cases aged 60–69, 79% of cases aged 70–79 and 51% of cases aged ≥80. The main treatment differences by age were the proportions having chemotherapy, ranging from 70% of cases aged 45–59 to 18% of cases aged ≥80. The lowest proportion with cancer-related treatment among all subgroups examined was 17% for cases with other/unspecified histology.

Fifty-one percent of cases died within one year of diagnosis (89% from lung cancer), 67% and 74% died within two and three years respectively; 5% of controls died within three years. The Kaplan-Meier estimate for 5-year lung cancer-specific survival was 27% and for overall

**Table 1. Characteristics of eligible lung cancer cases diagnosed 2006–2013, corresponding population proportions, proportions survived >1 year and "average case" costs.**

| Subgroup | No. of cases | % of cases | Population comparison (% of cases)[a] | % survived >1 year | "Average case" cost to 3 years[b] ($) |
|---|---|---|---|---|---|
| All cases | 994 | 100 | | 49 | 51,944 |
| Sex | | | *NSW 2010–2014* | | |
| Female | 432 | 43 | 42 | 56 | 54,784 |
| Male | 562 | 57 | 58 | 43 | 49,713 |
| Age at diagnosis (years) | | | *Australia 2007* | | |
| 45–59 | 125 | 13 | 16 | 54 | 67,689 |
| 60–69 | 305 | 31 | 27 | 56 | 63,513 |
| 70–79 | 342 | 34 | 33 | 50 | 51,026 |
| ≥80 | 222 | 22 | 23 | 32 | 29,541 |
| Tumour histology | | | *Australia 2007* | | |
| Small cell carcinoma | 102 | 10 | 12 | 34 | 51,057 |
| Non-small cell carcinoma | 721 | 73 | 63 | 51 | 55,368 |
| *Squamous cell* | *163* | *16* | *16* | *56* | *53,002* |
| *Adenocarcinoma* | *403* | *41* | *29* | *56* | *61,711* |
| *Large cell* | *155* | *16* | *17* | *34* | *42,603* |
| Other specified carcinoma | 124 | 12 | 15 | 52 | 41,125 |
| Other/Unspecified | 47 | 5 | 11 | 28 | 31,713 |
| Stage at diagnosis | | | *NSW 2010–2014* | | |
| Localised | 197 | 20 | 18 | 86 | 51,531 |
| Regional | 220 | 22 | 21 | 65 | 57,905 |
| Distant metastases | 481 | 48 | 46 | 26 | 54,543 |
| Unknown | 96 | 10 | 15 | 47 | 36,462 |
| Year of diagnosis | | | *Not applicable* | | |
| 2006–2008 | 143 | 14 | | 47 | 54,810 |
| 2009 | 176 | 18 | | 52 | 55,877 |
| 2010 | 178 | 18 | | 47 | 50,002 |
| 2011 | 163 | 16 | | 47 | 52,514 |
| 2012 | 173 | 17 | | 49 | 46,773 |
| 2013 | 161 | 16 | | 49 | 51,943 |
| Smoking status | | | *Not available* | | |
| Never / Ex quit >15 years | 461 | 46 | | 51 | 53,737 |
| Ex-smoker quit < = 15 years | 249 | 25 | | 47 | 54,046 |
| Current smoker | 284 | 29 | | 47 | 47,120 |
| Remoteness of place of residence | | | *NSW 2010–2014* | | |
| Major cities | 505 | 51 | 67 | 48 | 51,305 |
| Inner regional | 346 | 35 | 24 | 51 | 54,718 |
| Outer regional/(Very) Remote | 143 | 14 | 9 | 44 | 47,465 |
| Socioeconomic quintile[c] | | | *NSW 2010–2014* | | |
| Most disadvantaged quintile | 312 | 31 | 22 | 48 | 51,653 |
| Quintile 2 | 248 | 25 | 26 | 42 | 49,176 |
| Quintile 3 | 162 | 16 | 21 | 57 | 62,509 |
| Quintile 4 | 156 | 16 | 17 | 49 | 49,591 |
| Least disadvantaged quintile | 116 | 12 | 14 | 53 | 46,718 |
| Health insurance status[d] | | | *Not available* | | |
| Private insurance | 423 | 43 | | 54 | 56,991 |
| Concession card | 385 | 39 | | 44 | 47,591 |

*(Continued)*

**Table 1.** (Continued)

| Subgroup | No. of cases | % of cases | Population comparison (% of cases)[a] | % survived >1 year | "Average case" cost to 3 years[b] ($) |
|---|---|---|---|---|---|
| None | 186 | 19 | | 46 | 50,241 |

Totals may not add to 100% due to rounding. Costs in 2013 Australian dollars.

[a] Comparison with cases diagnosed in New South Wales 2010–2014 [36] or cases diagnosed in Australia 2007 [32].

[b] Costs from 1 year pre-diagnosis to 3 years post-diagnosis for cases diagnosed to June 2013 (n = 916).

[c] Based on place of residence at recruitment.

[d] Insurance status reported at recruitment.

5-year survival was 21%, compared with 5-year relative survival of 18% for all lung cancer cases diagnosed in Australia 2011–2015 [2]. Cases with other/unspecified histology, of whom 47% had distant metastases and 21% had unknown disease stage, had median survival of 2 months.

## "Average case" costs

Overall, the mean excess cost for each case from one year prior to until three years after diagnosis was $51,944. Observed costs were highest for younger cases ($67,689 for age 45–59), those from the middle socioeconomic quintile and cases with adenocarcinoma (Table 1). Costs were lowest for cases aged ≥80 ($29,541), and those with other/unspecified histology or unknown disease stage in the NSWCR.

From multivariable analysis of the independent effects of the key characteristics, average costs were strongly associated with age at diagnosis ($p<0.0001$), smoking status ($p<0.0001$), disease stage ($p = 0.002$) and socioeconomic group ($p = 0.01$) (Table C in S1 File). Higher costs were associated with younger age at diagnosis (versus lower costs for cases aged ≥80), while lower costs were associated with current smoking at study recruitment, having unknown disease stage in the NSWCR, and being from less disadvantaged socioeconomic areas. There was a suggestion of lower costs for people from outer regional and remote areas ($p = 0.05$) and those with "other specified carcinoma" histology (not SCLC or NSCLC; $p = 0.07$). There was no evidence of differences in costs by health insurance status ($p = 0.28$), year of diagnosis ($p = 0.58$) or sex ($p = 0.81$). Excluding cases with unknown stage, there was no significant variation in costs by disease stage ($p = 0.54$). Type of treatment (surgery, chemotherapy, radiotherapy) was strongly associated with costs but their inclusion in the multivariable regression did not greatly alter the associations between costs and the other factors. Cases with distant metastases had higher costs than those with unknown stage; 70% of metastatic cases had cancer-related treatment compared with 56% of those with unknown stage. Restricting the analyses to cases with three matched controls did not alter any associations examined. Restricting to cases who died within three years after diagnosis gave similar results, although the adjusted association between average costs and socioeconomic disadvantage was no longer statistically significant ($p = 0.11$).

## Monthly costs

Excess costs for cases were highest immediately after diagnosis (particularly in the first month) and in the final month of life for those who died (Figs 3 and 4). Excess costs >4 months prior to diagnosis were minimal (Table A in S1 File). Costs were predominantly for hospital inpatient services, accounting for 81% of the excess costs in the month after diagnosis, which decreased to ~60% from 5 months after diagnosis onwards (Table D in S1 File). The

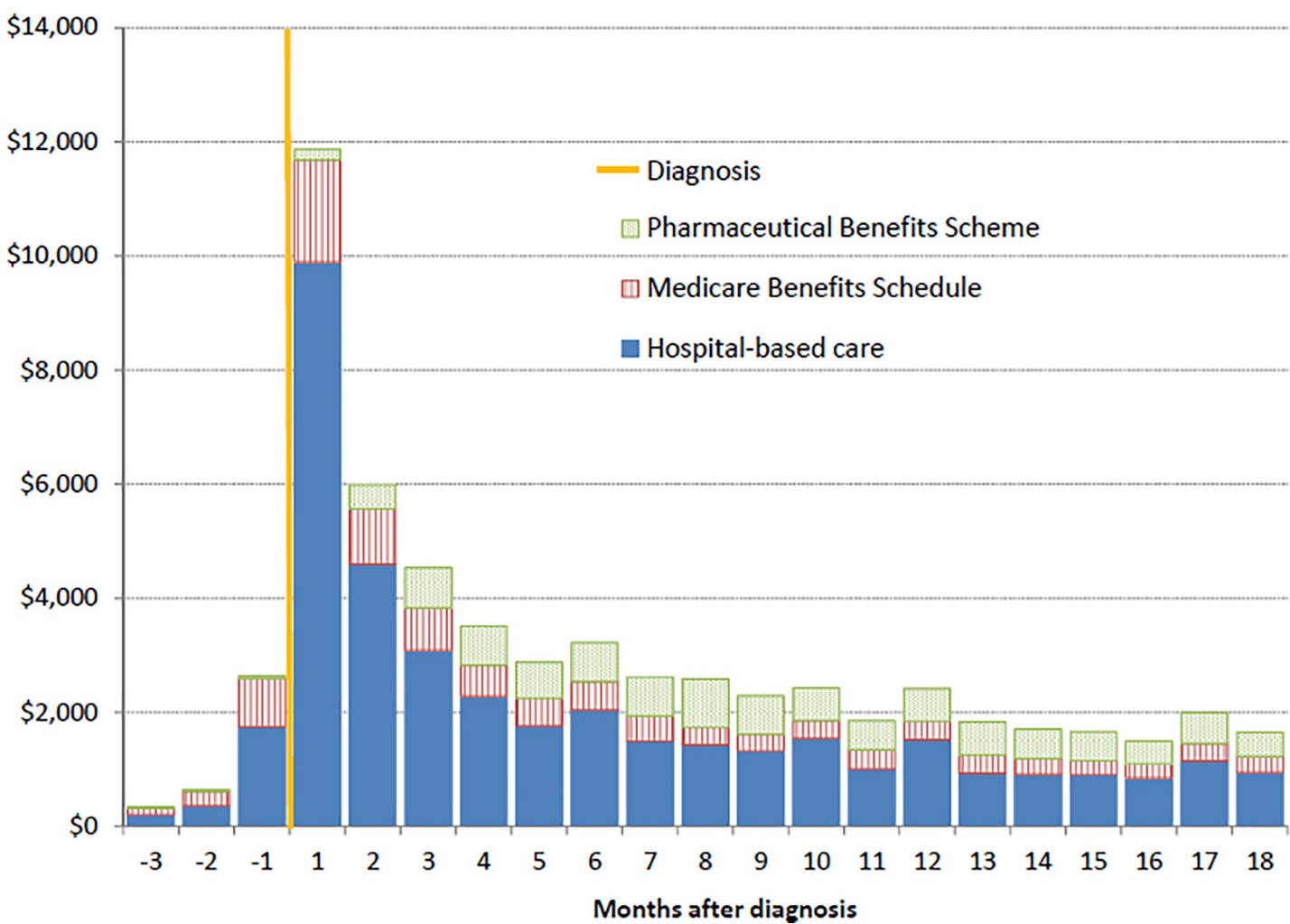

**Fig 3. Monthly excess costs by source, relative to diagnosis, for eligible incident lung cancer cases diagnosed 2006–2013.** In 2013 Australian dollars, for cases alive at the start of each month.

proportion of costs attributed to prescribed medicines increased from ~5% within two months of diagnosis to 20%-30% from 4 months after diagnosis onwards. Medical services covered by the MBS accounted for 30%-40% of costs in the 3 months prior to diagnosis (including chest and/or abdominal CT scans for 68% of cases) and then accounted for ~15% per month up to 18 months after diagnosis.

For the 79% of cases who died, there was a gradual increase in costs from ~6 months prior to death, with a large spike in excess costs to ~$15,000 during the final month of life (Fig 4). This pattern was similar regardless of length of survival, apart from slightly higher costs for those surviving <1 year after diagnosis. The sources of the excess costs for 6–12 months prior to death were 55–60% inpatient hospital costs, 20–25% prescription medicines and 15–20% other medical services covered by the MBS, with ED costs accounting for <5% of the excess costs in each month. During the final month of life, 89% of the excess costs were for inpatient hospital care and of the cases who died, 69% had a record of dying in hospital.

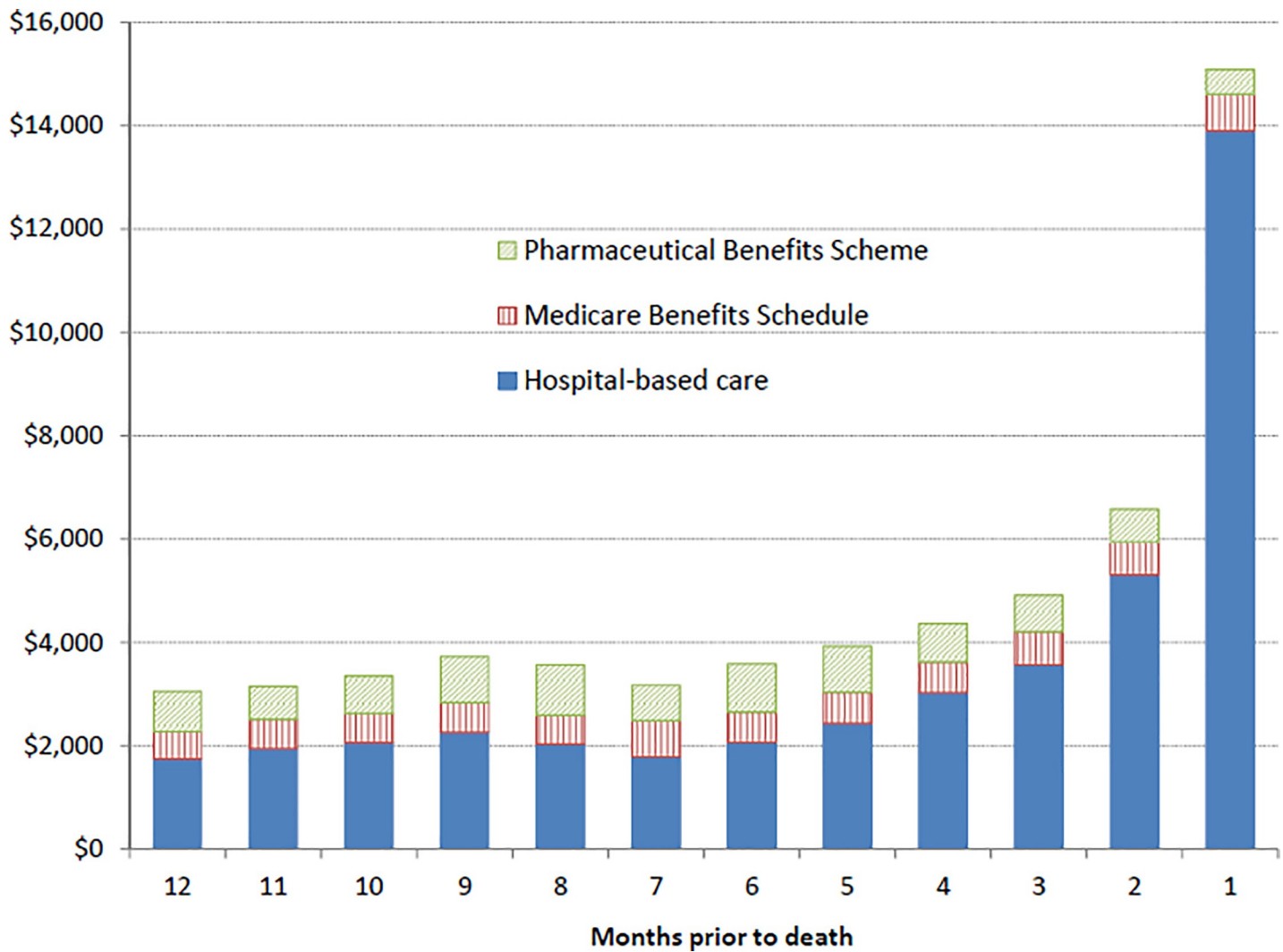

**Fig 4. Monthly excess costs by source at the end of life, for eligible incident lung cancer cases diagnosed 2006–2013 who died to June 2016.** In 2013 Australian dollars.

## Costs by phase of care

Excess costs during the initial and terminal phases were predominantly for inpatient care (Fig 5; Table D in S1 File). However, during the continuing care phase 54% of excess costs were for prescription medicines–the proportion was 58% for NSCLC cases and 32% for the other histological types. The higher costs for prescription medicines included those for drugs such as pemetrexed (cytotoxic chemotherapy, recorded for 13% of all 994 cases) and erlotinib (targeted therapy, on PBS from 2008, recorded for 9%). For cases who survived >5 years, the mean annual excess costs for years 3–5 after diagnosis consistently averaged ~$5,000. From multivariable analysis of the independent effects of the key characteristics, costs in the terminal phase were associated with age and stage (each $p<0.0001$), smoking status ($p = 0.003$) and insurance status ($p = 0.01$), with associations similar to those observed for average case costs (e.g. lower costs for elderly, unknown stage, smokers) and slightly higher costs for those with private health insurance. During the initial treatment phase, costs were similarly associated with stage ($p<0.0001$) and age ($p = 0.001$), but no other factor. In the continuing care phase,

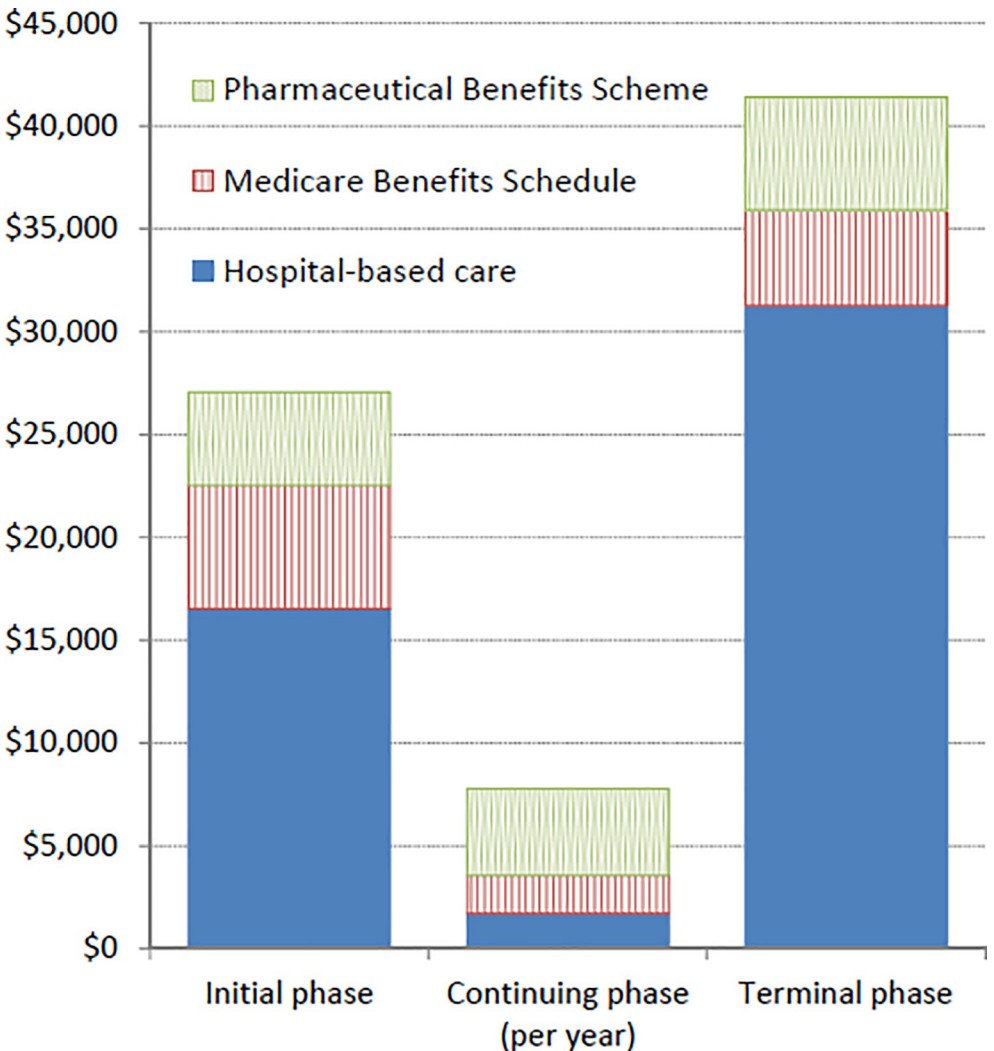

**Fig 5. Excess costs by phase of care and source, for eligible incident lung cancer cases diagnosed 2006–2013, in 2013 Australian dollars.**

costs were associated with stage ($p = 0.0001$), age ($p = 0.005$) and smoking status ($p = 0.005$), with higher costs for people with metastatic disease and cases aged 45–59 years and lower costs incurred for smokers. Tables E-G in Supporting information show the summary costs for each phase and treatment levels by histological type and stage.

## Exploratory analysis of cases diagnosed in 2014–15

Using hospitalisation data, we identified 465 lung cancer cases whose first diagnosis record was during 2014–2015. Of these, 29% had an MBS-subsidised EGFR molecular test (~$400 each, added to MBS May 2012, required for erlotinib eligibility from 2014) compared with 19% and 11% of the NSWCR matched cases diagnosed in 2013 and 2012 respectively (6% of all diagnosed 2006–2013). Gefitinib was the first targeted therapy for lung cancer subsidised in Australia (from 2004); treatment with this medication was recorded for 0–2% of cases each year for 2006–2015. Erlotinib (in PBS from 2008) was recorded for 9% of the 994 NSWCR matched cases diagnosed 2006–2013, including 15% of cases diagnosed 2008–2009 (median 10

months from diagnosis to first use) and 6–10% of cases diagnosed during 2010–2013, but only 3% of new cases diagnosed during 2014–2015 (median 1 month to first use). There were very few records of crizotinib or ALK testing for crizotinib eligibility (both on PBS from July 2015). Overall, 4% (n = 18) of cases diagnosed 2014–2015 had a subsidised targeted therapy and 29% had a subsidised molecular test. Eleven percent of those tested had a subsidised targeted therapy. No other relevant government-subsidised targeted therapies or molecular tests were added to the PBS or MBS during the study period.

## Discussion

To our knowledge this is the first study to provide detailed health system costs for lung cancer care in Australia, stratified according to various patient and tumour characteristics and using population-based estimates from individual-level data. This gives a unique insight into the factors associated with healthcare costs for lung cancer. These results provide key inputs for cost-effectiveness analyses of both existing and emerging lung cancer interventions, including lung cancer screening, tobacco control strategies, and new treatment technologies. We found substantial variation in costs of care for key patient and tumour subgroups. Costs tended to be lower for subgroups with shorter survival such as the very elderly and current smokers. There was also substantial variation in costs by phase of care, with higher costs during the initial treatment phase and the terminal phase. Costs were highest in the month following diagnosis and the month prior to death, largely attributable to inpatient hospital services, and were strongly associated with age at diagnosis. The costs in the continuing care phase were also substantial, with higher costs for subsidised prescription medicines and cases with metastatic disease.

Age at diagnosis was a key determinant of healthcare costs, with higher costs incurred for those who were younger at diagnosis. This suggests that, even after allowing for disease characteristics, younger cases were more likely to access more (and/or more expensive) health services. Younger cases more commonly had cancer-related treatment compared with older cases, particularly chemotherapy, suggesting they might have more robust health status and access to a wider range of treatment options, and/or that they are more likely to be selected for these treatments for other reasons. Conversely, there were lower costs for smokers, particularly in the terminal phase, possibly due to more comorbidities and poorer health, limiting intensive treatment options. There were also low observed (unadjusted) costs for people who had unknown histology and/or unknown disease stage recorded in the NSWCR. These cases generally had very short survival, with consequently limited opportunity to complete a full diagnostic work-up or receive extended treatment. Somewhat surprisingly, there was little difference in costs for cases with localised, regional, or distant stage. The only difference identified was higher costs in the continuing care phase for people with distant metastases at diagnosis, indicating more intensive ongoing treatment–this is an important factor to consider in relation to potential changes in disease stage distributions.

In our previous study of overall excess costs for all cancer types, we reported that costs for lung cancer were similar to the average for all cancer types, with slightly higher costs for the continuing care phase and ~10% lower costs for the terminal phase [4]. This more detailed stratified analysis indicated that a much higher proportion of costs were attributed to subsidised medicines for the continuing care phase (54%) relative to other periods. This was also higher than the costs for medicines during the continuing phase for all cancers combined (35%), partly due to the higher prevalence of advanced lung cancer, where systemic therapy is more frequently used. Costs for medical services covered by the MBS increased across the 3 months prior to diagnosis, as previously reported for all cancer types [4]. For lung cancer,

these costs are potentially due to poor health in advance of the diagnosis and/or diagnostic uncertainty. Lung cancer patients are often symptomatic for many months before diagnosis, with multiple symptoms that cannot be differentiated from other conditions; this frequently leads to a delay in their diagnosis [37]. Our previous work reported that one-third of all NSCLC patients presented to emergency departments around the time of their diagnosis, and these patients had a higher number of comorbid conditions [38].

Using a previously validated algorithm and hospital admission records, we were able to identify more recently diagnosed lung cancer cases in our study cohort [27, 28]. There was some uptake of EGFR tests for these cases but still relatively low use of subsidised targeted therapies for cases diagnosed to 2015. From 2014, a positive EGFR test was required to receive subsidised erlotinib and from this time there was a substantial increase in testing but a drop in erlotinib use. However, identifying cases this way does not capture all cases and histology/staging information was not available to allow more informed analyses of these cases. This will be an area for further investigation as more recent NSWCR data become available. However, these data are informative for marking the beginning of an upward trend in the use of new technologies and medicines in real-world lung cancer care. Our cost estimates included a small component of existing costs relating to molecular testing (6% of cases) and targeted therapy (10%). However, there is potential for substantial increases in costs, as a range of new targeted therapies/immunotherapies were added to the PBS (e.g. afatanib, ceritinib, nivolumab) from 2017, and some existing medicines (e.g. gefitinib) have been approved for a larger proportion of lung cancer cases [17]. Future estimates of costs will also need to include more widespread molecular/biomarker testing of lung cancer patients to determine their eligibility for targeted therapies. Also, future improvements in survival will potentially increase costs through additional ongoing treatment.

This study has several important strengths. We used empirical data from a large population-based sample with detailed individual-level data and included all cases without restriction to a specific hospital catchment or treatment types. We had information on specific items and procedures for individuals providing a better understanding of real-world patterns of care and costs. We matched cases with up to three controls to give more robust results. The cases were identified through a comprehensive high-quality state-wide cancer registry. Finally, the nature of the prospective 45 and Up Study cohort and linkage to routinely collected health datasets eliminated the need to rely on patients' recall of services accessed.

However, the lung cancer cases may not be representative of all cases in Australia. 45 and Up Study participants have been shown to be healthier and more likely to have private health insurance than the general population [39] and the proportion of lung cancer cases who were never smokers (15%) may be higher than for all lung cancer cases [40]. We excluded 2% of cases who were first diagnosed on their death certificate, as they could not have treatment after diagnosis, along with a small number of cases whose month of diagnosis was not recorded, as their healthcare use could not be accurately assigned to time periods relative to diagnosis. While we matched cancer-free controls on age and place of residence, the oversampling of people aged ≥80 and those from rural areas in the 45 and Up Study cohort may have some impact on our estimated costs, with cases from more disadvantaged areas potentially over-represented. However, we derived excess cancer costs by comparing individuals within the cohort and adjusted for key factors in regression analyses. A recent paper from the 45 and Up Study reported that study participants diagnosed with lung cancer had longer post-diagnosis survival, but the cohort had similar levels of hospital and ED use as the NSW population [41]. Furthermore, our cohort was comparable to the total population of lung cancer cases on most key factors. We weighted the cohort to match the population-level stage distribution and by

matching cancer cases with similar cancer-free controls we estimated relative differences in costs. Relative comparisons in the cohort have been shown to be unbiased [39].

The comparisons for "average case" costs used excess costs from one year before to three years after diagnosis. While this covers all costs for the majority of cases, we could not investigate in greater detail the subsequent healthcare use of the longer-surviving cases. Future work can include analysis beyond three years, particularly for continuing care costs, as end-of-life costs were similar regardless of length of survival, as has been shown elsewhere [14]. While these comparisons of "average cases" elucidate the differences between patient subgroups, they do not represent the actual costs incurred for individuals, as dollar amounts have been inflated or deflated to be standardised across calendar years allowing more direct comparisons. However, they do provide informative relative cost comparisons for lung cancer care across an important range of characteristics.

We could not account fully for the hospital-based costs of chemotherapy or radiotherapy delivered on an outpatient basis in public hospitals, however the costs of the drugs subsidised by the PBS were captured [4]. Not all EDs in NSW provided data to the EDDC for the entire study period. However, the EDDC covered >80% of NSW ED presentations in 2007 [42] and increased to near-complete coverage, so the vast majority of ED presentations were included. Furthermore, ED costs accounted for a very small proportion of all costs. We restricted our analysis to direct health system costs due to a lack of comprehensive data for other costs, such as patients' out-of-pocket costs. Importantly, treatment decisions (and the corresponding costs) are driven by patients' performance status and detailed staging information relating to clinical guidelines, along with patients' choice, but as this information was not available we could not assess the appropriateness of treatments.

In 2019, the Australian Government announced a major enquiry into the prospects and feasibility of implementing a targeted lung cancer screening program in Australia [24], which would involve a structured approach to early detection in high risk individuals. Our prior assessment of the cost-effectiveness of lung cancer screening using trial based eligibility criteria found that lung screening would not be cost-effective, but this was based on older cost data for lung cancer treatment (from 2005–2008) [22]. The more current costs of lung cancer treatment elucidated in the current analysis will provide key information for an updated evaluation of the benefits of lung cancer screening, since the increasing costs of therapy over time imply that early detection strategies are likely to be increasingly economically beneficial. Interestingly, we did not identify evidence for a major difference in costs by known stage at diagnosis, apart from in the continuing care phase. The economic benefits of lung screening will also depend on its potential to enhance survival and quality-adjusted life-years saved. Work to reassess the health and economic impact of lung cancer screening, using the cost data presented here, is ongoing. Our work program will enhance future activity in this area through the development of a comprehensive microsimulation modelling platform for lung cancer in Australia. This will include detailed estimated costs to assess the cost-effectiveness of current and emerging interventions, such as the introduction of new drugs, screening and early detection programs and primary prevention strategies.

## Conclusions

The costs of lung cancer care are substantial and are lower, on average, for people with features associated with poorer survival. Costs appeared stable by year of diagnosis during 2006–2013. A key aim of future health services planning for lung cancer is to improve survival, and it is important to know about the potential cost implications. These cost estimates provide important information for health services planning and delivery and the evaluation and

implementation of new interventions for cancer control. Costs are likely to grow with the wider availability of new therapies and technologies, and increasing survival times. These findings provide detailed recent data and a benchmark for future costs of lung cancer care; they will inform new estimates of the economic benefits of lung cancer screening.

## Supporting information

**S1 File.** Table A. Summary statistics for excess costs overall, by phase of care and by month for incident lung cancer cases diagnosed 2006–2013. * For cases alive at the start of each time period. Costs reported in Australian dollars for the year 2013. Table B. Codes for identifying lung cancer-related treatment with surgery, chemotherapy or radiotherapy*. * Records included were from one month prior to cancer registry diagnosis date onwards. APDC: Admitted Patient Data Collection; ATC: Anatomical Therapeutic Chemical Classification System; MBS: Medicare Benefits Schedule; PBS: Pharmaceutical Benefits Scheme. Table C. Multivariable analysis of costs for lung cancer cases diagnosed 2006–2013. * Excludes cases with missing values for any of the listed variables and any cost outliers. Results for each variable are adjusted for all listed variables. Table D. Summary mean excess costs for eligible incident lung cancer cases diagnosed 2006–2013, with proportion of the excess costs by source. * Hospital-based services combines Admitted Patient Data Collection and Emergency Department costs, the latter accounted for 1–3% of costs for each time period except in the 3 months before diagnosis (5–6%). Costs reported in 2013 Australian dollars. MBS: Medicare Benefits Schedule; PBS: Pharmaceutical Benefits Scheme. Table E. Summary excess costs by phase of care and selected tumour characteristics, for eligible incident lung cancer cases diagnosed 2006–2013. NSCLC: non-small cell lung cancer, includes "Other specified carcinoma"; SCLC: small cell lung cancer. * Spread of disease at diagnosis as reported by the New South Wales Cancer Registry. Some of these numbers are relatively small and estimates may be somewhat unstable. Table F. Summary excess costs by in the year prior to diagnosis, by selected tumour characteristics, for eligible incident lung cancer cases diagnosed 2006–2013. NSCLC: non-small cell lung cancer, includes "Other specified carcinoma"; SCLC: small cell lung cancer. * Spread of disease at diagnosis as reported by the New South Wales Cancer Registry. Table G. Number of cases in each phase and proportion having anti-cancer treatment, by selected tumour characteristics, for eligible incident lung cancer cases diagnosed 2006–2013. CI: confidence interval; NSCLC: non-small cell lung cancer, includes "Other specified carcinoma". [a] Anti-cancer treatment includes surgery, chemotherapy and radiotherapy as defined by Table B, from one month prior to cancer registry diagnosis date onwards. [b] Spread of disease at diagnosis as reported by the New South Wales Cancer Registry.
(DOCX)

## Acknowledgments

This research was completed using data collected through the 45 and Up Study (www.saxinstitute.org.au). The 45 and Up Study is managed by the Sax Institute in collaboration with major partner Cancer Council NSW; and partners: the National Heart Foundation of Australia (NSW Division); NSW Ministry of Health; NSW Government Family and Community Services–Ageing, Carers and the Disability Council NSW; and the Australian Red Cross Blood Service. We thank the many thousands of people participating in the 45 and Up Study, the Centre for Health Record Linkage for the record linkage and the Department of Human Services, the NSW Ministry of Health and Cancer Institute NSW for the use of their data.

## Author Contributions

**Conceptualization:** David E. Goldsbury, Marianne F. Weber, Sarsha Yap, Karen Canfell, Dianne L. O'Connell.

**Data curation:** David E. Goldsbury.

**Formal analysis:** David E. Goldsbury, Marianne F. Weber, Sarsha Yap.

**Methodology:** David E. Goldsbury, Marianne F. Weber, Sarsha Yap, Nicole M. Rankin, Preston Ngo, Lennert Veerman, Emily Banks, Karen Canfell, Dianne L. O'Connell.

**Writing – original draft:** David E. Goldsbury, Marianne F. Weber, Sarsha Yap.

**Writing – review & editing:** David E. Goldsbury, Marianne F. Weber, Sarsha Yap, Nicole M. Rankin, Preston Ngo, Lennert Veerman, Emily Banks, Karen Canfell, Dianne L. O'Connell.

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
