## [Decision Letter · Decision Letter 0]

26 May 2020

PONE-D-20-04316

Health services costs for lung cancer care in Australia: Estimates from the 45 and Up Study

PLOS ONE

Dear Dr. Goldsbury,

We are sorry about the delay because of waiting for some valuable comments. Please kindly respond to the reviewers' comments in detail.  

We look forward to receiving your revised manuscript.

Kind regards,

Jason Chia-Hsun Hsieh, M.D. Ph.D

Academic Editor

PLOS ONE

Additional Editor Comments (if provided):

We are sorry about the delay because of waiting for some valuable comments. Please kindly respond to the reviewers' comments in detail.

2. In your ethics statement in the manuscript and in the online submission form, please provide additional information about the patient records used in your retrospective study. Specifically, please ensure that you have discussed whether all data were fully anonymized before you accessed them.

3.  To comply with PLOS ONE submission guidelines, in your Methods section, please provide additional information regarding your statistical analyses. For more information on PLOS ONE's expectations for statistical reporting, please see https://journals.plos.org/plosone/s/submission-guidelines.#loc-statistical-reporting.

Reviewers' comments:

Reviewer's Responses to Questions

**Comments to the Author**

1. Is the manuscript technically sound, and do the data support the conclusions?

Reviewer #1: Partly

Reviewer #2: Yes

Reviewer #3: Yes

2. Has the statistical analysis been performed appropriately and rigorously? 

Reviewer #1: Yes

Reviewer #2: N/A

Reviewer #3: Yes

3. Have the authors made all data underlying the findings in their manuscript fully available?

Reviewer #1: No

Reviewer #2: No

Reviewer #3: Yes

4. Is the manuscript presented in an intelligible fashion and written in standard English?

Reviewer #1: Yes

Reviewer #2: Yes

Reviewer #3: Yes

5. Review Comments to the Author

Reviewer #1: Review

Health services costs for lung cancer care in Australia: Estimates from the 45 and Up

Study

Overview: In this descriptive study, the authors provided detailed population-based estimates of health system costs for lung cancer care, as a benchmark prior to wider availability of targeted therapies/immunotherapy.

Overall, the study was done with academic rigor, using a unique dataset. However, I note below several concerns:

Major comments:

1. The overarching concern is that the study appears to be insufficiently motivated, in the strict sense that the stated purpose cannot logically be fulfilled by the results of the study. The reasons are that the study justifies its primary aim with the following sentence: “Given the large number of new cases every year, information about changing costs for lung cancer treatment is crucial to health system planning in Australia and in other developed countries.”

a. First, there is no justification why Australia’s experience can guide health system planning in other developed countries. In fact, very little is given to highlight essential features of the Australian healthcare system, so that an international reader from another developed nation may not necessarily know how applicable lessons learned from this study would be to his or her national setting.

b. The primary innovation from the study is stated to be the subgroup analyses in terms of how costs are generated for cancer care in Australia. This sentence (1 above) does not clearly speak to the need for subgroup analyses.

c. Related to point (b) above, there was no explanation/justification as to why the specific subgroups were chosen for analyses. This justification should be tied into the motivation for the study.

d. In fact, lines 75-77 actually provide the information that I was hoping to find earlier on in the introduction. I suggest that the authors move this critical sentence higher up: “It is important to establish methods for estimating overall costs of treatment over time, and to benchmark tumour-specific treatment costs prior to the wider introduction of new targeted therapies and the accompanying molecular testing.” But then see point 2 below:

2. The second issue also relates to motivation. The manuscript states, in lines 75-77 (see 1d above), that we need to benchmark costs prior to wider introduction of new targeted therapies. However, the costs that are estimated and documented are actually costs prior to the new targeted therapies, so they have no direct relevance with the costs of the newer therapies. I find it difficult to understand the rationale for understanding the costs of “old technologies” before introducing “new technologies.” In fact, in line 83-84, the manuscript again notes “to understand future costs, we need to know the costs of key patient groups.” But I am not sure how knowing the costs of current technology for key patient groups will inform future costs, at least not without further explanation.

3. The third issue again relates to motivation: If the rationale for the study is to understand the costs of care prior to adopting new technology, why is it important to understand the excess costs due to lung cancer (relative to patients with no lung cancer)? Should we need to know just how much it costs to treat lung cancer; not “how much more” lung cancer costs relative to someone who is similar, but has no lung cancer? In fact, the decision to define “excess” costs also leads to downstream issues, such as those identified in lines 227-229 (the need to shift costs to avoid negative excess costs). If we simply used actual costs (not excess costs), there would be no need to account for negative excess costs.

4. There may be some confusion as to how “population-level” this study is. The abstract states that the study is “population-level,” but the methods section actually tells a different story – that information was gathered from a trial (that oversamples certain groups); and these patients were selected to participate in a study/survey. Lines 215-27 suggest that the study had to be re-weighted in order to match population-level data. So I would refrain to qualify this study as “population-level.”

5. In the study sample section (starting on line 130), there are certain exceptions that are self-evident; but other exclusions are not so clear. For example, why exclude keratinocyte cancers?

6. I am curious about the matching approach that includes matching based on smoking status, particularly matching based on current smoking status. If a case who has smoked for 20 years must be matched with a control who has smoked for 20 years without cancer, this would seem to introduce some odd selection issue.

7. Finally, I would like to reiterate the importance of establishing the importance of identifying certain subgroups to compare costs. As currently written, the article reads like a collection of facts. I understand that not all manuscripts are hypothesis-driven and therefore will not necessarily progress in a linear fashion. However, a descriptive study that presents a lot of results by slicing the data in different ways actually ends up being difficult to read. Perhaps motivating the reasons for looking at costs for different subgroups may help lessen this sense? I.e., assessing costs by cancer stage seems quite appropriate. But what about all the statistics presented under “average costs”? Or facts and figures under “monthly costs”? Are not monthly costs highly correlated with the stage of cancer? Also, the exploratory analyses of new cases from 2014-2015 seem ad hoc – They seem to be an afterthought because they are not sufficiently developed relative to the other sections.

8. Without ending on a morbid note, the final paragraph indirectly suggests that a cost-saving technique in cancer care is to diagnose later. I would suggest rewording the last paragraph, so that the paragraph does not begin with noting that “lower costs are associated with shorter survival time … and this is important information for health services planning.”

Minor comments:

1. What does it mean that ED costs were assigned based on triage category and discharge status? Why not just simply tally up all the costs during a specific ED visit?

2. Line 303 seems to be contradicted by the table: Isn’t the cost for current smoker lower? The text says costs for current smokers are higher.

Smoking status Not available

Never / Ex quit >15 years 461 46 51 53,737

Ex-smoker quit <=15 years 249 25 47 54,046

Current smoker 284 29 47 47,120

Reviewer #2: The manuscript relies on descriptive stats and simple mean comparisons of matched samples to analyze the ‘costs’ of lung cancer treatment in Australia. The authors find that as expected the cost of lung cancer treatment is lower for older patients, varies across different types of tumor histology, and is strongly inflationary. That is all rather trivial.

The authors also find an inverse u-shaped relation between socioeconomic status and treatment costs. Patients from most and least deprived socioeconomic strata are on average less costly. This is interesting, but the authors do explain these findings. One has to say that the manuscript remains unanalytical and simply describes differences without trying to explain them.

From a methodological perspective I do not believe that patient characteristics are uncorrelated. Therefore, most of the reported mean differences will be biased if the treatment and control group are not perfectly matched in all relevant dimensions. What is clearly missing, to give an example, are the existence of diseases other than lung cancer.

In sum, I find the manuscript borderline publishable. I also cannot see many implications. Yes, the costs of lung cancer treatments differ systematically, but so what?

I also find it implausible that the data cannot be made publicly available. This should be a reason to reject the paper. If data is not shared and no replication policy in place, it is not science.

Reviewer #3: This paper addresses an important issue of the Australian healthcare system. That of the lung cancer that was responsible for 9000 deaths in 2019 and accounts for a substantial proportion of Australian government healthcare costs. This shows us the seriousness of the problem in that country. The authors argued that It is important to establish methods for estimating overall costs of treatment over time and to benchmark tumor-specific treatment costs before the wider introduction of new targeted therapies and the accompanying molecular testing. However, there are little published Australian data quantifying these costs, and the authors proposed to provide detailed population-based estimates of the health system costs for lung cancer care using linked patient-level data from New South Wales (NSW), Australia.

The authors give a good and detailed explanation of their data sources data collection as well as how the major variables such as cost were measured and the different assumptions that were made in their calculation.

The authors’ findings indicate that excess costs from one-year pre-diagnosis to three years post-diagnosis averaged ~$51,900 per case. In addition, the costs were higher for cases diagnosed at age 45-59 ($67,700) or 60-69 ($63,500), and lower for cases aged ≥80 ($29,500) and those with unspecified histology ($31,700) or unknown stage ($36,500). They argued this is the first study to provide detailed health system costs for lung cancer 391 care in Australia.

The paper is well written, the data are well presented.

The analysis is well done and the discussion to me looks fine.

However, after knowing those different costs, one could ask the question of “so what?”

They may be the need to related these costs to the standard of living of those patients. From the Australian standard is it something that an average individual can afford or some measure need to be put in place to support patients? This is important to better inform policymaking.

6. PLOS authors have the option to publish the peer review history of their article (what does this mean?). If published, this will include your full peer review and any attached files.

Reviewer #1: No

Reviewer #2: No

Reviewer #3: Yes: Paul M. DONTSOP NGUEZET

---

## [Author Response · Author response to Decision Letter 0]

25 Jun 2020

The responses are in the attached file "Response to Reviewers.docx".

---

## [Decision Letter · Decision Letter 1]

10 Aug 2020

Health services costs for lung cancer care in Australia: Estimates from the 45 and Up Study

PONE-D-20-04316R1

Dear Dr. Goldsbury,

We’re pleased to inform you that your manuscript has been judged scientifically suitable for publication and will be formally accepted for publication once it meets all outstanding technical requirements.

Kind regards,

Jason Chia-Hsun Hsieh, M.D. Ph.D

Academic Editor

PLOS ONE

Additional Editor Comments (optional):

Reviewers' comments:

Reviewer's Responses to Questions

**Comments to the Author**

1. If the authors have adequately addressed your comments raised in a previous round of review and you feel that this manuscript is now acceptable for publication, you may indicate that here to bypass the “Comments to the Author” section, enter your conflict of interest statement in the “Confidential to Editor” section, and submit your "Accept" recommendation.

Reviewer #3: All comments have been addressed

2. Is the manuscript technically sound, and do the data support the conclusions?

Reviewer #3: Yes

3. Has the statistical analysis been performed appropriately and rigorously? 

Reviewer #3: Yes

4. Have the authors made all data underlying the findings in their manuscript fully available?

Reviewer #3: Yes

5. Is the manuscript presented in an intelligible fashion and written in standard English?

Reviewer #3: Yes

6. Review Comments to the Author

Reviewer #3: The authors have addressed properly my concerns about the first draft of the manuscript. I do not have further questions on the manuscript and i think the manuscript can now be accepted for publication in PlosOne journal.

7. PLOS authors have the option to publish the peer review history of their article (what does this mean?). If published, this will include your full peer review and any attached files.

Reviewer #3: No

---

## [Editor Report · Acceptance letter]

18 Aug 2020

PONE-D-20-04316R1 

Health services costs for lung cancer care in Australia: Estimates from the 45 and Up Study 

Dear Dr. Goldsbury:

I'm pleased to inform you that your manuscript has been deemed suitable for publication in PLOS ONE. Congratulations! Your manuscript is now with our production department. 

Kind regards, 

on behalf of

Dr. Jason Chia-Hsun Hsieh 

Academic Editor

PLOS ONE